# Study of Polyaniline/Poly(Sodium 4-Styrenesulfonate) Composite Deposits Using an Electrochemical Quartz Crystal Microbalance for the Modification of a Commercial Anion Exchange Membrane

**DOI:** 10.3390/membranes10120387

**Published:** 2020-11-30

**Authors:** Antonio Montes-Rojas, Marlen Ramírez-Orizaga, Jesús Gerardo Ávila-Rodríguez, Luz María Torres-Rodríguez

**Affiliations:** Laboratorio de Electroquímica, Facultad de Ciencias Químicas, Universidad Autónoma de San Luis Potosí, Av. Dr. Manuel Nava, Zona Universitaria, San Luis Potosí CP. 78210, Mexico; marlenram@gmail.com (M.R.-O.); avila_rdz@hotmail.com (J.G.Á.-R.); luzmaria@uaslp.mx (L.M.T.-R.)

**Keywords:** poly(sodium 4-styrenesulfonate), polyaniline, anion exchange membrane, electrochemical quartz crystal microbalance

## Abstract

One of the intended applications for the modification of ion exchange membranes with polyaniline (PAni) is to use it as a matrix to include chemical species that confer a special property such as resistance to fouling or ion selectivity. In particular, the inclusion of polyelectrolyte molecules into the PAni matrix appears to be the way to modulate these properties of selective membranes. Therefore, it must be clearly understood how the polyelectrolyte is incorporated into the matrix of polyaniline. Among the results obtained in this paper using poly(sodium 4-styrenesulfonate) (PSS) and an electrochemical quartz crystal microbalance, the amount of polyelectrolyte incorporated into PAni is found to be proportional to the PSS concentration in solution if its value is between 0 and 20 mM, while it reaches a maximum value when the PSS in solution is greater than 20 mM. When the anion exchange membranes are modified with these composite deposits, the transport number of chloride was found to decrease progressively (when the PSS concentration in solution is between 0 and 20 mM) to reach a practically constant value when a concentration of PSS greater than 20 mM was used.

## 1. Introduction

Among the most important characteristics that ion exchange membranes must have, for a successful application, are adequate selectivity and/or resistance to biofueling, for example, for ions of similar charge and size or for bacteria to be present in the solution [1]. In the literature, several different approaches exist to handle these disadvantages that are a feature of membranes [1,2,3,4,5,6]. For example, one approach is to produce a novel membrane with suitable selective properties; however, this is a complex multistep process [7,8,9]. Another approach consists of modifying a characteristic of a commercial membrane [4,5,6]. This method is less expensive, since the membranes conserve most of their transport properties and only the properties related to selectivity or resistance to fouling change.

For this last approach, there have been several reports on the modification of ion exchange membranes using polyaniline. Polyaniline (PAni) is considered to be the most promising material modifier due to its excellent properties, easy synthesis, low cost of the monomer, and high mechanical and thermal stability as compared to other conducting polymers [10,11].

Some studies have reported that the membranes might be modified superficially by PAni [12,13] or with impregnations or intercalations on the walls of their microchannels [14,15,16,17,18,19]. In both cases, the PAni generates special transport properties at ion exchange membranes that are completely understood.

For example, Compañ et al. and Berezina et al. [16,19] studied cationic heterogeneous membranes modified with polyaniline intercalations and concluded that the doping of polyaniline macromolecules imparts electrical heterogeneity to the membrane surface, which is due to the development of a microrelief. This microrelief may affect the coupled effects of concentration polarization and optimizes the structure and properties of composite membranes.

In another case, a superficial film of PAni on the membrane was found to work as a positive barrier that repels positive ions and improves its selectivity. For example, it was found [12,13,19,20] that these composite membranes have lower values of electroosmotic permeability as compared to those of the initial membrane.

Despite the importance of the effects of these modifications to polyaniline on the ion exchange membranes, another possibility is that the polyaniline deposited in the membrane functions as a matrix that includes chemical species that confer some special property such as resistance to fouling or selectivity improvements. Regarding the latter, if it is considered that the interactions between ionic species and the superficial layer or the doped PAni enhances the selectivity, then the strength of these interactions can be used to moderate the selectivity of the composite membrane. In our laboratory, we have proposed an approach to control the strength of the ion-PAni interactions on an ion exchange membrane: To deposit on the membrane, a layer or impregnations of PAni with inclusions of a polyelectrolyte such as sodium polystyrene sulfonate are used. Due to the long chain lengths of the polyelectrolytes, they are trapped inside the conducting polymer film, leading to the formation of polymer composites. The positive charges introduced into the polymer matrix during oxidation of the conducting polymer are usually counterbalanced by the anions present in the molecule of polyelectrolyte. In the literature, there are many articles on this issue that have tried to understand these composites using a metallic electrode [21,22,23] such as gold, but they have not been used to modify a membrane.

For this approach, it is necessary to rigorously control diverse parameters such as the morphology and redox state of polyaniline and the amount of polyelectrolyte included. These characteristics of the PAni composite can be controlled using an electrochemical method in their preparation, because these methods are simple and easy to implement. In addition, they consist of only a few stages. In accordance with the literature, when using metallic electrodes, the role of the polyelectrolyte (PE) in composite deposits is to neutralize the charge of PAni due to doping. This charge compensation process involves the molecules of PE aligning opposite to charges of PAni [24,25]. On the other hand, the literature has [26,27,28] reported that the PE affects the morphology and the amount of PAni deposited onto the electrode.

Importantly, a valuable tool in this regard is the electrochemical quartz crystal microbalance (EQCM). The principle of EQCM and several examples of its use are summarized in references [29] and [30]. In the field of conducting polymers, the EQCM was introduced by Orata and Buttry [31], who showed that the doping process and the redox state of PAni are strongly related. Thus, the deprotonation of amine groups in the PAni (leucoemeraldine base) occurs during its oxidation (electron loss), which produces the imine group (emeraldine base and pernigraniline base). These forms can in turn be protonated, and thus charged sites were formed in the material. To maintain the electroneutrality of the PAni, anions of small molecular sizes are usually exchanged with the solution during the doping-dedoping processes of the conducting polymer. Specifically, the variation in the mass involved with the exchange of ions was detected by EQCM.

More recently, Ding and Park [32] studied the incorporation of PSS (poly(sodium 4-styrenesulfonate)) into a deposit of PAni using the EQCM. They also analyzed the electropolymerization of aniline onto a gold electrode with and without PSS in solution. Their results showed that in both cases, the charge associated with PAni was the same, but the mass detected by EQCM was higher when PSS was in solution. This result is contrary to the conclusions of Hyodo et al. [25] because they found that the addition of PSS accelerated the growth rate of PAni. In addition, these researchers found that the PAni/PSS composite that formed had dominantly anion-exchanging characteristics. This property dominates at lower pH values, and its contribution decreases with an increase in the pH. This result is surprising and is not in agreement with what has been generally believed for this type of composite polymer [33,34,35].

However, it is not possible to find research with similar results related to this property of PAni/PSS composites. Only Pile et al. [36] analyzed the effect of oxidation state on the permeability of several probe molecules through conducting polymer membranes comprised of PAni composites in a PAni/PSS aqueous solution. For example, they found that the permeability of the reduced state of composite is very similar to that observed for the pure PAni film compared to the oxidized state. This behavior reflects the competition between the nature of the reduced film, which is more compact but is a pure cation exchanger with a fixed PSS charge, and the oxidized film, which is swollen but is not a pure cation exchanger, since it contains both negative charges from bound PSS and positive charges from the charge injected during oxidation.

Despite this information regarding the properties of PAni/PSS composites, their effect on the properties of a commercial membrane has not yet been studied. In addition, it is not even known what effect the amount of PSS has on PAni electropolymerization.

Considering the aforementioned, the effect of different amounts of polyelectrolyte on electropolymerization of PAni using an EQCM was analyzed for the first time in this study. Afterwards, the results obtained for the first step were extended to modification of a commercial anion exchange membrane with PAni/PSS composites.

## 2. Materials and Methods

### 2.1. Electromicrogravimetric Study of the PAni/PSS Composite Deposits

#### 2.1.1. Materials

The solutions used in this work were prepared with deionized water, with a resistivity of 18 MΩ cm obtained from a Mili-Q Reference Ultrapure Water Purification System, and aniline of reagent grade (Sigma-Aldrich, Saint Louis, MO, USA) and doubly distilled. The polyelectrolyte poly(sodium 4-styrenesulfonate) (PSS, Mw 70000, Acros Organics, Bridgewater, NJ, USA) was used as received.

The working solution had 0.1 M H_2_SO_4_ and 0.10 M aniline with PSS 2.5−40 mM. The electropolymerization of composite deposits was performed using 40 cyclic voltammetry scans with the interval potential limited by −100 mV and +1000 mV. The scan rate used was 50 mV s^−1^.

Finally, a 0.5 M solution of camphorsulfonic acid (HCS, Sigma-Aldrich, Saint Louis, MO, USA) was used to study the charge compensation process of the composites by using EQCM.

#### 2.1.2. Experimental Setup

All EQCM measurements were performed with a made-home electrochemical cell to facilitate the replacement of the resonators. An AT-cut, 9 MHz gold-plated quartz crystal (Seiko EG&G Au, Seiko EG&G Co., Ltd., Tokyo, Japan, 0.196 cm^2^) mirror finish was used as a resonator and an electrode. A platinum wire functioned as an auxiliary electrode and an Ag/AgCl/3 M KCl cell was employed as the reference electrode. Both electrodes were contained in a glass extension that only contained an acidic solution (0.1 M H_2_SO_4_) without aniline.

Both the quartz-crystal analyzer and potentiostat/galvanostat were obtained from Princeton Applied Research; the first was a Seiko EG&G model 917 and the second was an EG&G model 273A. The data acquisition was carried out using the software WinEchem 1.5 (Seiko EG&G Co., Ltd., Tokyo, Japan) from the same provider. A schematic representation of the experimental setup with the main components used in this determination is shown in Scheme 1.

#### 2.1.3. Characterization of the PAni/PSS Composite Films Using Scanning Electron Microscopy

Micrographs of the PAni/PSS composite films were obtained using Scanning Electron Microscopy (SEM), and the percentage of sulfur content (%S) by Energy Dispersive X ray (EDX) was also obtained. These analyses were performed with a QUANTA 200 FEI microscope (Thermo Fisher Scientific, Bridgewater, NJ, USA).

The handling of the samples was as follows: the quartz electrodes used to prepare the PAni/PSS composite deposits were placed directly on the sample holder. They were then placed under the electron beam to make the determinations.

### 2.2. Modification of the Commercial Anion Exchange Membrane

#### 2.2.1. The Commercial Membrane

The commercial anion exchange membrane was provided by Tokuyama Soda Co. Ltd. Tokio, Japan. This membrane is denominated as AFN and is a homogeneous membrane with NR_3_^+^ groups according to the manufacturer. Some characteristic properties of the AFN membrane are reported in Table 1. 

#### 2.2.2. Procedure for the Modification of Membranes

A carbon paste electrode was used as a support to adhere to the samples of the commercial membrane and proceed to their modification by using cyclic voltammetry, as described elsewhere [39]. Membrane discs with an area of 4.9 cm^2^ were used in this work. To remove any impurities in the membrane, the disc was treated as follows: initially, the membranes were immersed in 1 M HNO_3_ for at least 24 h with moderate agitation. Subsequently, the membranes were rinsed thoroughly using an ultrasound bath and Milli-Q water to remove any trace of nitric acid. Finally, the membranes were kept in 1 M H_2_SO_4_ for 24 h or longer until they were used.

### 2.3. Characterization of the Modified Membranes

#### 2.3.1. Electrochemical Characterization of Modified Membranes

After the membranes were modified with the different types of PAni/PSS composite deposits, each of them was reattached to the electrode and their voltammetric responses were then obtained using an aniline-free sulfuric acid solution.

#### 2.3.2. Determination of the Transport Numbers Using the Modified Membranes

The characterization of the modified membranes included the determination of the counterion transport number, t_a_, which was measured using the concentration cell method developed by Ltief et al. [40]. This method consists of using reference electrodes to measure the membrane potential (E_m_) generated when the membrane separates two solutions with different chemical activities.

The experimental setup used in this determination involved placing the membrane (with an exposed area to solutions of 0.7853 cm^2^) between two symmetric glass beakers connected with a 1 cm length and 1 cm internal diameter tube. Two Ag|AgCl|NaCl (3 M) reference electrodes were used to measure E_m_. Each electrode was placed inside a glass extension filled with 3 M NaCl. The top of the glass extension was blocked with an agar plug (3% *w*/*w*), which prevented loss of the NaCl solution and established the ionic contact. The membrane potential was measured every 10 s for 15 min with a UNI-T UT70C multimeter, which was connected to a computer by a R232 port and was controlled through the UT70C software, provided by the supplier. The experiments were performed under constant agitation, and the initial and final temperatures were recorded in each experiment. Furthermore, each experiment was performed at least three times.

For determination of the counterion transport number at a concentration of 0.01 M, it was employed a set of four solutions of different concentrations (namely, two higher and two lower concentrations); the counterion transport number was determined from the measured membrane potential according to the following equation:(1)$$E_{m}=\left(2t_{a}+1\right)\frac{RT}{zF}\ln\frac{a_{1}}{a_{2}}$$
where *a*_1_ is the activity of the constant concentration solution, and *a*_2_ is the activity of the concentration variable solution. It should be noted that these values were determined using the Guntelberg equation [41]. In addition, *R* represents the gas constant (8.314 J mol^−1^ K^−1^), and *T* represents the temperature (K), while *z* and *F* represent the counterion charge and the Faraday constant (96,500 C mol^−1^), respectively.

Finally, when working with the modified membranes, special care was taken to ensure that the modified side of the membrane was in contact with the higher concentration solution.

## 3. Results

### 3.1. Potentiodynamic Behavior of the PAni/PSS Composites on the Gold Electrode

The cyclic voltammograms recorded in sulfuric acid with the PAni/PSS composites obtained with different concentrations of PSS during their preparation are shown in Figure 1a. All the curves show two redox pairs with two oxidation peaks at approximately 300 mV and 700 mV and two reduction peaks at 0 mV and 450 mV. These characteristics correspond to the typical footprint of PAni in an acid medium [42,43], but do not correspond to that of the polyelectrolyte, since it is not electroactive.

According to different studies [42,43,44,45], the peaks mentioned above are attributed to redox processes of PAni that involve two stages of charge transfer. In the first, at 300 mV, the transformation of leucoemeraldine to the protonated emeraldine form occurs. The second stage, at 700 mV, involved the incomplete transformation of emeraldine to pernigraniline.

Further analysis shows that the peak potential is shifted towards high potential only for the oxidation peak at 700 mV with an increasing concentration of PSS, while for the other peaks, the increasing the concentration of PSS does not influence the position of their peak potentials. In addition, when the PSS concentration is less than 20 mM, the detected current increases proportionally with the increasing PSS concentration (curves B−E). This implies that increasing the concentration of polyelectrolyte present during the electropolymerization of PAni catalyzes its polymerization, which implies that the addition of PSS accelerated the growth rate on the gold electrode. Some studies [25,32] have shown that large amounts of PSS molecules are adsorbed on the positively polarized surface of the gold electrode during the first cycle of electropolymerization and may also retain small dimeric and oligomeric species. The PSS molecules adsorbed on the positively charged electrode surface may now act as a template for the polymerization of aniline, which not only helps accumulate weight quickly on the electrode surface but also facilitates the electron-transfer reaction across the polymer film. Likewise, it was observed that the peak current (I_p_), Figure 1b, such as the oxidation peak at 350 mV, increases with an increasing PSS concentration (curves B−E). Indeed, the current increases from 100 μA to 400 μA with an increasing PSS concentration. However, if the PSS concentration is above 20 mM (curves F and G), then the peak current decreases with an increasing PSS concentration. This behavior is best appreciated by examining Figure 1b.

Another parameter analyzed from voltammograms was the difference of potential between the anodic peaks, ΔE_p_. Indeed, the separation between anodic peaks tends to decrease with the addition of sulfonic groups to PAni. The corresponding values obtained for ΔE_p_ for each PSS concentration are shown in Figure 1c. Accordingly, ΔE_p_ decreases when the PSS concentration increases between 0 and 10 mM (curves B−D). Thus, the increasing incorporation of PSS into the PAni deposit decreases ΔE_p_ if the PSS concentration is between 0 and 10 mM. At a PSS concentration greater than 10 mM (curves E−G), the opposite effect is observed, i.e., an increase in PSS concentration leads to an increase in the ΔE_p_ value. This indicates that a maximum amount of PSS incorporates into the PAni matrix. Importantly, the behavior of ΔE_p_ is similar to the behavior of I_p_, since in both cases, an increase in the PSS concentration in solution leads to increases in the incorporation of PSS into the PAni matrix. However, if the concentration continues to increase, then there is less incorporation.

### 3.2. Electromicrogravimetric Behavior of PAni/PSS Composites on a Gold Electrode

During the electrochemical synthesis of PAni/PSS composites, the change in frequency (Δf = f − f_o_ in Hz) was measured, which was transformed into mass change (Δm) per surface unit by employing Sauerbrey’s equation:(2)$$\Delta f=-\frac{f_{o}{}^{2}}{{N\rho }_{q}}\Delta m$$
where the following terms are constant: f_o_ (fundamental frequency, Hz), ρ_q_ (quartz density), and N (constant related to the speed of the acoustic wave traveling through the quartz). These terms can be grouped in a constant named the sensitivity factor (C_f_) that is obtained by calibration, for example with silver, copper, or thallium [46,47,48].

According to this equation, the mass change can be obtained from the frequency change if C_f_ is known. The minus sign in this equation indicates that a decrease in the frequency implies an increase in the mass.

Figure 2 shows the frequency change recorded during the electrochemical synthesis of PAni films without polyelectrolyte in the work solution.

This result clearly indicates that the tendency of Δf is to decrease when the number of scan cycles increase, which is associated with the progressive growth of the PAni deposit on the gold electrode [32]. Additionally, a careful analysis of this graph shows that the frequency change is related to the doping-dedoping process, such as the decrease in frequency recorded from the last two cycles of potential scan between -100 mV and 600 mV. The same trend was observed in the frequency shift curves recorded during the doping-dedoping process for all PAni/PSS films.

In order to find further information on the effect of PSS concentration on the composite deposition growth, the mass change was obtained from the frequency shift curves recorded during the potentiodynamic polymerization of aniline with and without PSS in the work solution. Figure 3a shows the mass (Δm) of the PAni/PSS deposits generated as a function of the potential cycle number. From these curves, two important differences could be noted when the polyelectrolyte was added (curves B−G) or not (curve A) to the synthesis solution. The first one is related to the trend data; indeed, in absence of PSS (curve A), the mass values follow a linear trend with the potential cycle number. Meanwhile, in the presence of PSS, all curves (B−G) are not completely linear with a unique slope but exhibit two slopes. This characteristic implies that the growth of composite deposits was affected by the PSS concentration in the synthesis solution, and thus the polyelectrolyte was involved in the polymerization of the aniline.

The second difference is associated with the amount of PAni deposited in presence of different PSS concentrations to the same potential cycle number, since as can be seen from the Figure 3a (for example at 10 or 20 cycles), in presence of PSS, the amount of PAni deposited on the electrode is always higher than in the absence of polyelectrolyte. In addition, m has different behaviors depending on the PSS concentration.

In order to better appreciate these behaviors, Δm at 20 cycles depending on PSS concentration is shown in Figure 3b. Notably, from this curve, if the PSS concentration is less than 20 mM, then the mass change (the amount of PAni deposited on electrode) increases proportionally with the amount of polyelectrolyte used in the synthesis of PAni. In this case, the PSS accelerates the growth rate of PAni, as has been mentioned by several authors [25]. Conversely, if the PSS concentration is greater than 20 mM, then the amount of PAni deposited on electrode decreases with the increasing concentration of polyelectrolyte. To explain this behavior, it is necessary to consider the role of the polyelectrolyte molecules.

As has been mentioned in other works [24,25], the polyelectrolyte molecules are adsorbed on the positively polarized surface during the first cycle to obtain polyaniline so that they retain small dimeric and oligomeric species. Because the sulfonate groups lose their proton around pH 0.5–1.0, the polyelectrolyte molecule has a net electrical charge counterbalanced by counterions existing in the solution. Since the synthesis solution has a pH near 1, the polyelectrolyte molecules adopt an extended chain conformation due to the repulsion between sulfonic groups. In these conditions, almost 50% of the –SO_3_H^+^ groups from the added PSS are also in solution, and the neutral molecules have not lost their protons. However, the charged molecules are precisely those that adsorb onto the electrode and accelerate the growth rate, since they prevent small dimeric and oligomeric species from diffusing away from the electrode surface into the solution. Considering this, if the PSS concentration increases, then the charged flexible chains adopt a more compact conformation with loops and tails, due to the slight increase in ionic strength associated with the addition of PSS. These conformations are less effective in preventing the diffusion of the oligomeric species; given that increasing the PSS concentration also increases the amount of compact species (consequently decreasing the elongated species), these species decrease the growth rate of PAni.

Additionally, the PAni grows on the adsorbed polyelectrolyte, incorporating into its matrix charged molecules of PSS, meaning the mass difference detected in the presence or absence of PSS is quite important due to its exclusively catalytic effect.

As part of efforts to generate more information on this issue, measurements of the shift frequency were made without polarization (on an open circuit) of the electrode using a solution of 1 M H_2_SO_4_, which was added to 20 mM or 40 mM of PSS. These measurements were performed before and after adding PSS to the acidic solution. Figure 4 shows the obtained results.

The data obtained show that adding only 1 mL of 1 M H_2_SO_4_ at 5 min (dashed line) causes the frequency variations to remain almost invariant. However, if a 1 M H_2_SO_4_ solution is added to 20 mM or 40 mM PSS at a time of 10 min (see arrow), then the frequency change decreases (mass increases) to a practically constant value near −100 Hz.

This frequency change behavior suggests that the PSS effectively adsorbs on the electrode surface and improves the polymerization of the aniline, as was mentioned above.

### 3.3. Electromicrogravimetric Analysis of the Effect of Polyelectrolyte Concentration on the Electrosynthesis of Composite PAni/PSS Deposits

#### Analysis of the Mass/Charge Ratio at Different Concentrations of PSS

With the purpose of better understanding how the PSS concentration affects the polymerization of polyaniline, the mass variation (Δm) as a function of charge variation (ΔQ) was determined from the cyclic voltammograms and frequency shifts simultaneously recorded during the growth of polyaniline on the Au electrode, as shown in Figure 5.

As can be seen, the curve obtained in the absence of PSS (curve A in Figure 5a) shows a linear behavior, which implies that the mass recorded is directly proportional to the electroactive species deposited on the electrode. However, if the polyaniline was synthesized in the presence of PSS, then the curves (curves B−G) showed an inflection point that divides the curve into two segments with different slopes: At the start of the polymerization, for example in curve B, the segment is linear with a slope of 0.3 μg mC^−1^, but when the electrosynthesis advances the slope of the segment becomes lower. This last situation suggests that PSS incorporation into the PAni film decreases as the PAni deposit thickens.

The result is consistent with other reports [3] in which it was shown that as the thickness of PAni deposits increases, the amount of polyelectrolyte decreases. The electrostatic interactions exerted between polyelectrolytes chains could possibly affect the incorporation of PSS into PAni deposit.

Figure 5b shows the slope of the curves at the start of the polymerization as a function of PSS concentration in the work solution. The curve shows that the slope and PSS concentration increase proportionally at low PSS concentrations. If PSS concentration is between 10 and 20 mm, then the slope reaches a maximum value. Finally, when the PSS concentration is greater than 20 mM, then the slope decreases. This behavior of the slope suggests that when the concentration of polyelectrolyte in solution is low (down to about 20 mM), the quantity of PSS incorporated into the PAni deposits is directly proportional to its concentration in solution. In addition, if the PSS concentration in solution is greater than 20 mM, then the PSS incorporation into the PAni deposit is indirectly proportional to its concentration in solution. The origin of these results is still unclear, but could be related to a limitation on the amount of PSS that can be incorporated into the PAni deposits while the amount of PAni increases. Notably, a similar trend was reported in other studies [10] in which the incorporation of PSS into the PAni/PSS system was evaluated by infrared, but the authors did not explain their results.

### 3.4. Study of the Charge Compensation Process of the Composites

The evaluation of the charge compensation process for each of the PAni/PSS deposits prepared can provide information about how the polyelectrolyte is incorporated into the PAni deposit at different PSS concentrations in the work solution. The charge compensation process of a conductive polymer is a phenomenon that involves different chemical forms of the polymer that differ in their electronic, electrical, magnetic, optical, and structural properties. For example, when the polymer is partially oxidized by electrochemical anodic oxidation, then the removal of electrons from the backbone of the polymer is accompanied by a delocalization of positive charge across its entire polymeric chain (backbone π system). In this process, counter ions are introduced from solution, which stabilize (compensate) the charge on the polymer backbone. If the polymer is reduced, then the positive charge disappears, the polymer returns to its neutral state, and the counter ions leave the polymer. Since this process is similar to that present in semiconductors, it is also known as the “doping/dedoping process” and the counter ions as the “dopant”. Doping is a reversible process so that the original polymer can be reproduced with little or no degradation of the polymer backbone. Both the doping and dedoping processes involve dopant counter ions that stabilize the doped state.

When the compensation charge process of a doped polymer is followed by EQCM, then the mass of polymer increases due to the entry of counter ions into the polymer from solution. However, if the doped polymer passes to the dedoped state the mass decreases because the counter ions are expelled from the polymer.

If by some means the dopant ion already exists in the polymer before the doping/dedoping process, then the process of mass gain and loss can be used, in this case, to initiate the incorporation of PSS into PAni deposit.

In order to analyze the doping/dedoping process of the PAni/PSS composites, they were used in two 0.5 M acid solutions with anions of different sizes: sulfuric acid, H_2_SO_4_, and camphor sulfonic acid (HCS) with molar masses of 97.1 and 231.3 g mol^−1^, respectively. The doping/dedoping process was studied by using cyclic voltammetry coupled to an EQCM, and the potential was swept between −100 and 1000 mV using a scan rate of 50 mV s^−1^.

Figure 6 shows only the voltammograms and frequency shifts concurrently recorded with CVs during the doping/dedoping process for the PAni/PSS composites using the PSS concentration 0, 20, and 40 mM in 0.5 M H_2_SO_4_ and HCS 0.5 M.

The potentiodynamic curves (I−E) have the same characteristics mentioned in the paragraphs above and shown in Figure 1. As for the frequency shifts curves, all the responses show the same general form, i.e., the frequency shifts adopt a constant value in a specific potential range, at the beginning and end of the cycle, where no process occurs. While in the middle of the cycle, the frequency shifts adopt different values. For example, as shown in Figure 6b, when H_2_SO_4_ was used, the frequency reached a constant value of about −100 Hz between 600 mV and 1000 mV and reached about 250 Hz between 300 mV and −100 mV. This evolution of frequency shift is associated with the doping-dedoping process, which involves the anions of the solution and the charges of the polyelectrolyte present in the PAni deposit. When the PAni is oxidized, positive charges are introduced into the polymer matrix, and as a result, anions must necessarily migrate from the solution to neutralize the excess positive change. Consequently, the frequency shifts decreases from 250 Hz to −100 Hz (the total frequency shift, Δf(H_2_SO_4_), is almost 350 Hz). Importantly, in this process, the charges of the incorporated polyelectrolyte are also involved. In this way, both the anions and the polyelectrolyte neutralize the excess positive change injected into the polymer. In the case where HCS is used, the role of the polyelectrolyte in neutralizing the excess positive charge is more important because the large acid anion cannot enter the matrix. In this case, the frequency shifts decrease from 50 Hz to 16−110 Hz (almost Δf(HCS) = 160 Hz). Interestingly, the remarkable hysteresis between the frequency shifts is associated with the incorporation of counterions and their rejection from the matrix, which involves the large anion size of HCS.

According to the results obtained using H_2_SO_4_, the total frequency shifts values when using H_2_SO_4_ are always superior to those obtained using HCS, although the HCS anion is heavier than that of H_2_SO_4_, which means that in this electrolytic medium the anion plays a more important role in neutralizing the excess positive charges in the PAni than when using HCS. These results also indicate that the incorporated PSS plays a more important role in neutralizing the excess positive charges in the polymer than the ions coming from the solution, when HCS is used.

With the purpose of evaluating the relationship between the total frequency shifts obtained with the two anions, Δf(H_2_SO_4_) and Δf(HCS), a curve was prepared with Δf(H_2_SO_4_)/Δf(HCS) as a function of the PSS concentration, as shown in Figure 7.

According to these data, if the PSS concentration is less than 20 mM, then Δf(HCS) is almost half of Δf(H_2_SO_4_) and Δf(H_2_SO_4_)/Δf(HCS) is between 2 and 2.5. This implies that when the PSS concentration increases in the synthesis solution, more PSS is incorporated in the matrix of PAni and less of the HCS anion participates in the doping process. On the contrary, if the PSS concentration in the work solution is greater than 20 mM, Δf(HCS) increases and Δf(H_2_SO_4_)/Δf(HCS) decreases from 2.5 to 2. This implies that when the PSS concentration is greater than 20 mM in the synthesis solution, less PSS is incorporated into the PAni matrix and more of the HCS anion participates in the doping process.

In agreement with the results presented in the Figure 3b and Figure 5b, the amount of PE incorporated into the PAni matrix reaches a maximum value when the PSS concentration in the work solution is about 20 mM.

This behavior is probably related to the fact that the polyelectrolyte is complete dissociated if the concentration of PE in solution is less than 20 mM. In that state, the negatively charged molecule facilitates the polymerization of the aniline either as a doping agent of the polymer, formed during the previous potential scanning, or through its adsorption. In this concentration range, the amount of polyelectrolyte incorporated into the PAni matrix is proportional to the concentration of PSS in solution.

On the other hand, if the PSS concentration in solution is higher than 20 mM, then the dissociation of the polyelectrolyte decreases and some of PE does not dissociate and remains as neutral molecules. These species neither can participate in improving polymerization performance nor be integrated into the PAni matrix.

Notably, the amount of undissociated molecules increases as the concentration of PSS in solution increases.

### 3.5. Analysis of the Composite Deposits by Scanning Electron Microscopy

To better understand the effect of the PSS concentration in solution on the morphology and the composition of the composites, micrographs were obtained using scanning electron microscopy (SEM), and the percentage sulfur content (%S) by Environment Dispersing X ray (EDX) was obtained.

Figure 8 shows the images of PAni deposits obtained using different PSS concentrations in the working solution. These images clearly show the influence of PSS concentration during the formation of the polymer.

Effectively, the PAni deposit obtained in absence of PSS (Figure 8a) shows a smooth surface that implies a fine porosity or a granular PAni film. The morphological characteristics of these films evolve with the PSS content in solution. As for the composite deposits obtained using 5 and 10 mM of PSS, shown in Figure 8b,c, their surfaces are characterized by small pores that are homogeneously distributed. When the PSS concentration increases to 20 mM (Figure 8d), then a surface composed of small well-defined grains is observed. Finally, the deposit obtained using a 40 mM PSS concentration, shown in Figure 8e, shows large perfect spherical clusters (see for example marked section) that can combine to form both short and long filaments.

First, the deposit of PAni in the absence of PSS is granular, as has been reported when using sulfuric acid as a working solution [49].

Second, as already mentioned above, the pKa of PSSH (poly(4-styrenesulfonic acid)) is approximately 1.0 [50,51]. Thus, when the PSS concentration in the work solution is low, almost 50% of the sodium styrenesulfonate groups in PSSH are converted to electrically neutral styrenesulfonic acid groups. Due to the small number of negatively charged sites on the adsorbed polyelectrolyte in their salt form, polymerization of aniline resulted in PAni polymers that are loosely anchored on the polyelectrolyte template, which results in small PAni/PSS composite particles. The PAni in these deposits has small grains regularly distributed on the surface, as shown in Figure 8b−d. When the PSS concentration in solution increases, more styrenesulfonate groups on the polyelectrolyte remain in their salt form. Consequently, polymerization of aniline results in shorter PAni polymer chains that stick tighter on the polyelectrolyte, producing smaller PAni/PSS composite nanoparticles. These deposits are characterized by small grains that can increase in size and combine to form short and long filaments.

Accordingly, the polyelectrolyte must gradually integrate into the PAni deposits. Figure 9 shows the content of sulfur in the deposits (%S) as a function of PSS concentration in work solution [52].

As can be seen, the amount of polyelectrolyte incorporated is actually proportional to the PSS concentration between 0 and 20 mM, while it reaches a maximum value when the PSS in solution is greater than 20 mM. If the PSS concentration in solution is higher than 20 mM, then %S decreases slightly.

This behavior of the content of sulfur, %S, is the same as that observed with the amount adsorbed of PSS detected by EQCM (see Figure 4); therefore, the reasons for this should be the maximization of electrostatic repulsions between previously adsorbed polyelectrolyte molecules, as discussed above.

### 3.6. Modifications of the Commercial Anion Exchange Membranes with PAni/PSS Composites

To better understand whether the behavior of the results obtained in the previous sections extends to those obtained using commercial membranes, the membrane was modified with PAni in presence of different amounts of PSS in the working solution. Figure 10 shows the voltammetric responses obtained for the deposits on the membranes in 1 M H_2_SO_4_ without aniline.

First, in Figure 10, all the responses are observed to have three pairs of peaks, denoted I, II and III that are characteristic of the typical response of PAni in acidic solution [42,43].

Additionally, a more rigorous analysis shows that these responses follow a behavior similar to that described in Section 3.1. To appreciate this behavior, the peak current as a function of the PSS concentration in the work solution of the anodic peak I was found, as shown in Figure 11.

Clearly, this curve exhibits the same behavior as that of the curve in Figure 1b with an inflection point at 20 mM, which indicates that the PSS has the same effect on the PAni deposits obtained on the solid electrode as on the membrane. This implies not only that the electrochemical properties of the composite deposit, discussed in previous sections, are not affected by the membrane, but also that the transport properties of the membranes may be affected by these types of deposits.

For example, Figure 12 shows the transport numbers of Cl^−^ obtained using these modified membranes.

According to these data, the transport number of Cl^−^ decreases by increasing the concentration of PSS between 0 and 20 mM. If the concentration of PSS is increased from 20 to 40, then the transport number of Cl^−^ slightly increases. It is important to note the remarkable relationship between this curve and the amount of PE incorporated in the deposits of PAni shown by the Figure 9.

Effectively, the transport number of Cl^−^ decreases due to the increase in the incorporation of PE into the PAni matrix when the PSS concentration in solution is between 0 and 20 mM. When even more PE is incorporated into the deposit when the PSS concentration increases, the Cl- anion will find increasing opposition to its passage through the membrane, because of the electrostatic repulsions between the PE and Cl^−^. Therefore, the transport number will decrease. In the case where the PSS concentration in solution is 40 mM, then the transport number increases slightly because of the small decrease in the amount of PSS incorporated into the PAni matrix.

## 4. Conclusions

In this paper, the effect of the concentration of poly(sodium 4-styrene sulfonate) (PSS) in solution on the electropolymerization of polyaniline (PAni) in acid solution using a EQCM was analyzed for the first time. Then, the results obtained from first step are extended to the modification of a commercial anion exchange membrane with these PAni/PSS composites.

Importantly, when the PSS concentration in solution is between 0 to 20 mM, then the amount of PSS that incorporates into the PAni matrix was found to be directly proportional to its value in solution. For PSS concentrations in solution above 20 mM, the amount of polyelectrolyte incorporated into the PAni matrix was found to reach a practically constant value.

The use of the EQCM to analyze the effect of the incorporation of PSS into the PAni matrix showed that the polyelectrolyte acts in the following two ways:

(a) The polyelectrolyte stimulates the polymerization of the PAni if its concentration in solution is between 0 and 20 mM by adsorbing onto the electrode surface and avoiding the diffusion towards the solution of the oligomers.

(b) By studying the doping-dedoping process, we found that the PSS incorporated into the PAni matrix could play an important role in neutralizing the excess positive charge in the doped polymer if a voluminous counterion such as the anion of HCS (camphorsulfonate) is in solution.

Finally, regarding the modification of a commercial anion exchange membrane with composite deposits, several results were found, which were in agreement with those obtained using the solid electrode. For example, the transport number of chloride decreases progressively (when the PSS concentration in solution is between 0 and 20 mM) to reach a practically constant value when a concentration of PSS that is greater than 20 mM was used. This result shows that when the amount of PSS increases in the composite then the electrostatic repulsions with approaching anions are more intense, resulting in a smaller transport number.

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
