# Peer review of "Study of Polyaniline/Poly(Sodium 4-Styrenesulfonate) Composite Deposits Using an Electrochemical Quartz Crystal Microbalance for the Modification of a Commercial Anion Exchange Membrane"

_membranes, 2020, doi:10.3390/membranes10120387_

Round 1

Reviewer 1 Report

Dear Authors!

The manuscript seems to be interesting for the researchers of the electrochemistry field and, in my opinion, may be published in the Journal. 

Nevertheless, I recommend you to fix some points in the article.

  1. Some indices are missed, for example "0.1 M H2SO4". 
  2. The quality of figures has to be improved because Figs. 4, 5b, 7, 9, 11, 12 in the current version of the manuscript have poor resolution.
  3. EDX analytical method acronym, as well-known, means "Energy-dispersive X-ray spectroscopy" but not "Environment Dispersing X ray".
  4. It is also questionable whether the method can determine the concentration of sulfur atoms at several percent (Fig. 9). It seems to me reasonable to give references to literature data with examples.

Author Response

Dear reviewer:

  1. Reviewed
  2. Almost all the figures were re-prepared.
  3. Done.
  4. Reference 52 was added to support the result.

I appreciate your comments.

Reviewer 2 Report

The manuscript presents the effect of different amounts of PSS on electropolymerization of PAni using an EQCM study. Then, the obtained results are extended to modification of a commercial anion exchange membrane with PAni/PSS composites. The work is well structured and described. I judge to be useful and important to the field and I recommend it to be published.

Minor points:

  • in Figure 1a, only 2 curves (F and G) are clearly indicated and I have the impression that one of them is missing because I can't identify 5 curves from A to E. I recommend assigning a precise identification to each curve, as it was done for F and G.
  • Figure 3a: I recommend assigning a precise identification to each curve, as it was done for F and G.

Author Response

Dear reviewer

  1. Done.
  2. Done.

I appreciate your comments.

Reviewer 3 Report

  1. Though the manuscript is well written, I am not quite sure what is the novelty of this study.
  2. One major limitation of this paper is the use of very old references. Most of them are >10 years. Out of 50, only around 7 are from 2015-2017, nothing beyond 2018 until the present, and most are <2010. This goes to show that the discussions are not based on the latest literature.
  3. Both abstract and conclusions can be made succinct.
  4. The EQCM schematic should be supplied as this is the main method used in the study.
  5. Add additional details for characterization in Sec 2.1.3.
  6. Sec 2.2.2: Give further details of the modification procedure.
  7. 3 is not characterization of the modified membrane. It is more of the concentration cell test. Please use appropriate section/sub-section headings.
  8. 3.1 – This section is a direct copy paste of your previous paper (DOI: 10.1039/C7RA03331A). Make sure to modify this.
  9. L174-176: Add corresponding units for the terms in the equation.
  10. Make Figure 1 bigger. Especially Fig 1a, the lines need to be clearer. Only F and G are shown on the plot. Where are A-E?
  11. L186-188: Delete these lines from “This section may….that can be drawn.” Looks like this is an instruction rather than a discussion.
  12. Fig 2: Make this bigger and clearer!
  13. Fig 3a: label the figure properly!
  14. L454: perfect spherical clusters. Where are your referring in Fig 8e?
  15. Figure 8 is not clear at all.
